# [RE] An Implementation of Fair Robust Learning

## Reproducibility Summary

**Scope of Reproducibility**

This work attempts to reproduce the results of the 2021 ICML paper To be Robust or to be Fair: Towards Fairness in Adversarial Training. I first reproduce classwise accuracy and robustness discrepancies resulting from adversarial training, and then implement the authors' proposed Fair Robust Learning (FRL) algorithms for correcting this bias.

**Methodology**

In the spirit of education and public accessibility, this work attempts to replicate the results of the paper from first principles using Google Colab resources. To account for the limitations imposed by Colab, a much smaller model and dataset are used. All results can be replicated in approximately 10 GPU hours, within the usual timeout window of an active Colab session. Serialization is also built into the example notebooks in the case of crashes to prevent too much loss, and serialized models are also included in the repository to allow others to explore the results without having to run hours of code.

**Results**

This work finds that (1) adversarial training does in fact lead to classwise performance discrepancies not only in standard error (accuracy) but also in attack robustness, (2) these discrepancies exacerbate existing biases in the model, (3) upweighting the standard and robust errors of poorly performing classes during training decreased this discrepancy for both both the standard error and robustness and (4) increasing the attack margin for poorly performing classes during training also decreased these discrepancies, at the cost of some performance. (1) (2) and (3) match the conclusions of the original paper, while (4) deviated in that it was unsuccessful in helping increasing the robustness the most poorly perfmoring classes. Because the model and datsets used were totally different from the original paper's, it is hard to to quantify the exact similarity of our results. Conceptually however, I find very similar conclusions.

**What was easy**

It was easy to identify the unfairness resulting from existing adversarial training methods and implement the authors' FRL (reweight) and FRL (remargin) approaches for combating this bias. The algorithm and training approaches are well outlined in the original paper, and are relatively accessible even for those with little experience in adversarial training.

**What was difficult**

Because of the resource limitations imposed, I was unable to successfully implement the suggested training process using the authors' specific model and dataset. Also, even with a smaller model and dataset it was difficult to thoroughly tune the hyperparameters of the model and algorithm.

**Communication with original authors**

I did not have contact with the authors during the process of this reproduction. I reached out for feedback once I had a draft of the report, but did not hear back.

# 1 Introduction

The advent of adversarial examples (1)(2) has motivated the need for procedures which decrease the sensitivity to noise of learned models (which I will call adversarial robustness or simply robustness.) Once such method is adversarial training (3)(4), in which adversarial examples are generated during the training process and are mixed in with "clean" examples to create mixed training batches of both manipulated and unmanipulated images. Learning on these batches has been shown to improve the robustness of models to adversarial attacks, often at a slight cost to standard performance (accuracy.)

To be Robust or to be Fair: Towards Fairness in Adversarial Training identifies that adversarial training creates unfairness in the resulting robust model. While the overall robustness of the model improves, some classes in the resulting model are more robust to adversarial attacks than others. Not only are the robustness benefits unfairly distributed, so too are the standard performance losses; the classes which are less robust at the end of the procedure tend to be the ones which suffer more in terms of standard performance. Moreover, these classes tend to be the ones which were harder to learn before adversarial training. As Xu et al. describe it: "adversarial training tends to make the hard classes even harder to be classified or robustly classified."

Motivated by this unfairness, Xu et al. conduct a theoretical analysis of the problem to explain this empirically observed phenomenon. They then draw on (5) to describe robust error in terms of the sum of standard errors (i.e. the probability that a class will be incorrectly classified without manipulation) and boundary errors (i.e. the probability that there exists some $\epsilon$-ball attack which can change a classifier's decision on a given class.) Using this description, they reformulate the learning problem into a series of cost-sensitive classification problems that can be penalized for violating fairness constraints. With this reformulation, they present two FRL algorithms for making adversarial training more fair: one which upweights the error of classes which violate the fairness constraints during training, and one which increases the attack radius for classes which violate fairness constraints during training.

# 2 Scope of reproducibility

The focus of this reproduction will be attempting to demonstrate the following:

- Claim 1, which is supported by Experiment 1 in Figure 1, is that adversarial training creates unfair outcomes in terms of both robustness and standard error.

- Claim 2, which is also supported by Experiment 1 in Figure 1, is that this unfairness exacerbates existing biases in model performance.

- Claim 3, which is supported by Experiment 2 in Figure 2, is that upweighting the error of classes which violates fairness constraints (using the authors' FRL: reweight algorithm) can improve the both the standard errors for the most poorly performing classes, and to a lesser degree their robustness.

- Claim 4, which is explored by Experiment 3 in Figure 3, is that increasing the margin of attack for classes which violates fairness constraints (using the authors' FRL: remargin algorithm) can also improve the fairness of the model– perhaps more effectively than reweighting.

# 3 Methodology

As an educational exercise, I aimed to re-implement the authors' training approaches from their descriptions in the paper. Because of the limitation imposed on the resources, however, I opted to use a simpler model and dataset in my experiments.

## 3.1 Model descriptions

The paper used the PreAct-ResNet18 and WRN28 architectures for their experimentation; I opted for the LeNet-5 architecture in the interest of efficiency. Though it is a much simpler model than the paper's originals, it provided enough complexity to conduct my experiments.

## 3.2 Datasets

The paper used the CIFAR10 and SVHN datasets for their experimentation; I used the Fashion-MNIST dataset. The train set is comprised of 60,000 examples, the test set 10,000. Both have a uniform label distribution across all 10 classes. The original train and test sets are used Experiment 1, while Experiments 2 and 3 split the train set into an 80/20 train/validation set for the FRL process. The only preprocessing done was to resize the images from 28x28 to 32x32. The data is freely available here.

## 3.3 Hyperparameters

The fairness tolerance hyperparameter was selected based on the recommendations in the paper (5%), as was the baseline $\epsilon$ (8/255 for the PGD attack.) For Experiment 1 I used a learning rate of 1e-3 for regular training and adversarial training, as the paper recommended. Due to resource constraints I had to limit the number of epochs I trained for to 15, and from convergence behavior I decayed the learning rate more often than the original paper (every 4 rounds by a factor of 3, as opposed to every 40 rounds by a factor of 10.) For the simpler model and dataset, this worked well.

For Experiments 2 and 3 I used a baseline learning rate of 1e-4, which I selected based on unstable behavior at a rate of 1e-3. I suspect this is due to differences in the model and dataset used, as well as the way I implemented the reweighting and remargining systems.

I utilized the results of the fairness evaluation ($\phi$ values) in the training process by applying a Softmax function to creating cross-entropy loss weightings, and as such the $\alpha$ values were different than the original paper's. I tried a variety of $\alpha$ values in the space of (1, 2, 5, 10,) and a variety of ratios of natural-$\alpha$s to boundary-$\alpha$s. The best results came from a ratio of 5:1 natural:boundary error weighting, which decreased the worst-case standard error by 25%, and the worst case robust error by 11%.

## 3.4 Experimental setup and code

For Experiment 1, I defined the LeNet-5 architecture and trained a classifier on the Fashion-MNIST dataset for 15 epochs at a learning rate of 1e-3. I then adversarially trained a new LeNet-5 model using a PDG attack for the same number of epochs at the same learning rate, with a 50/50 mixture of clean and manipulated images. I then compared the classwise standard accuracy (i.e. ability to predict a "clean" image correctly) and robust accuracy (i.e. ability to predict a image correctly despite manipulation) of the natural model and adversarially trained model. The results are recorded in Figure 1.

For Experiments 2, I retrained the unfair adversarially-trained model under the FRL (reweight) paradigm. During this procedure, I recorded the overall and classwise standard and boundary errors of the model during each batch, and based on these errors I re-calculated loss weights for each class. The loss function used was the sum of the standard loss and the loss for adversarially manipulated images with respect to the predictions on their unmanipulated counterparts (corresponding to standard error and boundary error, respectively.) Classes were penalized based on violations of fairness constraints, i.e. how greatly they differed from the average standard and boundary errors for all classes. I ran 10 rounds of retraining, and then compared the original unfair adversarially trained model with its retrained counterpart, comparing classwise standard and robust accuracy. These results can be found in Figure 2.

Experiment 3 was much the same as Experiment 2, the only difference being that instead of simply upweighting the loss of classes which violated fairness constraints, the radius of a class' attack during training was increased or decreased based on the size of their violation. Again, I ran 10 rounds of retraining, and then compared the original unfair adversarially trained model with its retrained counterpart, comparing classwise standard and robust accuracy. These results can be found in Figure 3.

All the code for these experiments, as well as example notebooks that walk through the procedure, can be found here.

## 3.5 Computational requirements

As mentioned, I used Google Colab for all of the experimentation. As such, it is difficult to describe the exact hardware that was used, or to even be confident of the consistency of the hardware throughout this process. I did use GPU resources, though I cannot speak to any specific type.

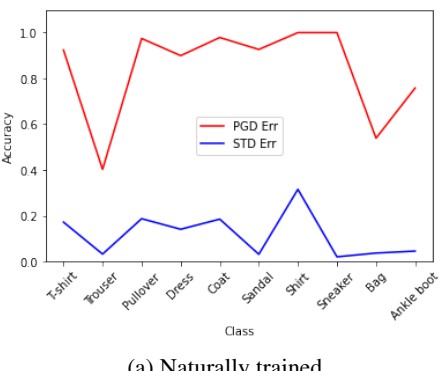
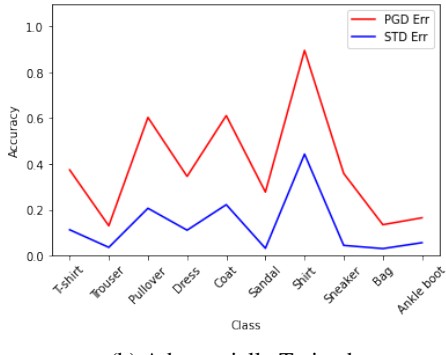

(a) Naturally trained                             (b) Adversarially Trained

Figure 1: Adversarial training produces unfair outcomes across classes, and worsens existing performance discrepancies

Experiment 1 can be run in approximately 15 minutes of GPU time. Experiment 2 can be run in approximately 5 hours of GPU time (for all alpha-combinations) and Experiment 3 can be run in approximately 3 hours.

All three notebooks can sometimes be run in parallel, but not always. Colab can be a bit unpredictable.

## 4 Results

In my experiments, I found that:

- Adversarial training does in fact lead to classwise discrepancies in standard error and adversarial robustness, that the least robust classes in the resulting model are the ones the model originally had a hard time learning, and that the penalties to standard performance brought on by adversarial training exacerbate existing biases in model performance.

- Reweighting the natural and boundary errors to penalize classes violating fairness constraints during adversarial retraining can improve the fairness of the model with respect to standard error, and to a lesser degree robust error.

- Remargining the attack radius for classes violating fairness constraints during adversarial retraining can also improve the fairness (i.e. lower the variance across classes) of the model's robustness (at a cost to robust performance) as well as improve the standard error.

Most of these results agree with the paper's conclusions, although the results in Experiment 3 differ in that I was not able to improve the robustness of the model with remargining as well as I could with reweighting. The original paper showed the opposite: that reweighting was unable to improve robustness for the most poorly performing classes. One experiment I did not conduct was to try both reweighting and remargining together, which the authors suggest might be fruitful. I leave that as a further exercise.

### 4.1 Results reproducing original paper

#### 4.1.1 Result 1

The result of Experiment 1 (shown in Figure 1) relates to claims 1 and 2 in Section 2. I found that in the naturally trained model, the standard error is quite low and the adversarial error (PGD error) is quite high. The adversarial error is not quite as uniform as in the original paper, I suspect because of the simplicity of the dataset and model I used.

Still it is observable that after adversarial training, the model's adversarial error is much lower across the board, but not in a fair way. Certain classes are much more robust to attack than others, and in particular the classes which had poorer initial standard performance are the ones with worse adversarial robustness. Moreover, we can see that there are penalties to standard performance incurred as a result of adversarial training, and the classes which suffer the most are the ones the natural model already had a hard time learning.

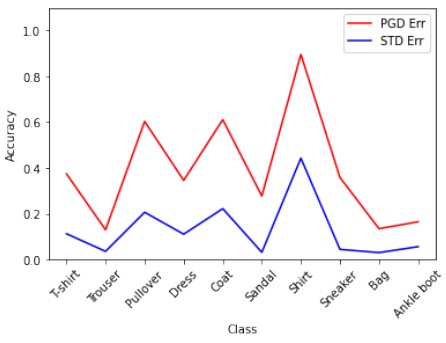 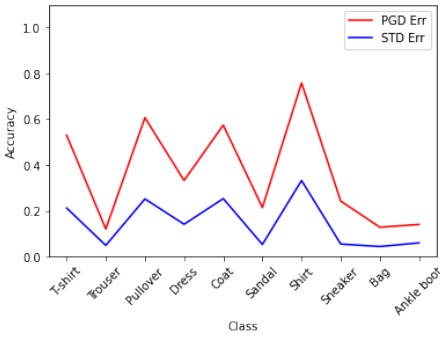

(a) "Vanilla" adversarially-trained model      (b) After the FRL (Reweight) procedure

Figure 2: FRL Reweight is able to mitigate standard performance losses, while also increasing the robustness of the most difficult class

Indeed, as Xu et al. put it, "adversarial training tends to make the hard classes even harder to be classified or robustly classified." This is exactly what I found, even with a totally different model and dataset.

### 4.1.2 Result 2

The result of Experiment 2 (shown in Figure 2) relates to claim 3 in Section 2. Here we can see the result of my best attempt at reweighting the loss of classes during adversarial retraining based on their violation of fairness constraints. As per the paper's FRL retraining algorithm, I began with an adversarially trained model and iteratively tried to retrain it, adjusting the loss of each class as I went depending on whether it violated fairness, and to what degree. As such, I compared the "vanilla" adversarially trained model with the resulting model after retraining.

I observed that for the hardest class to classify, there is a 25% reduction in standard error (bringing it nearly in line with the naturally trained model) and an 11% reduction in robust error. This is not totally free; we can observe, for example, that the standard and robust error for some of the easier classes suffers as a result. Still, the resulting model is fairer than it originally was.

These results seem relatively in-line with the original paper's, though again because of the different model and dataset selected it is hard to quantify the exact similarity. The overall conclusion is much the same though: reweighting is hugely successful in decreasing the classwise standard error discrepancies brought on by adversarial training, and to a lesser degree in decreasing classwise robustness discrepancies.

### 4.1.3 Result 3

The result of Experiment 3 (shown in Figure 3) relates to claim 4 in Section 2. This is the result of my best attempt at remargining during the retraining procedure. I observed a slight improvement in the worst-case standard error, but little to no improvement in the worst case robustness, and indeed a general degradation in robustness across most classes.

These results were not in line with the paper's, which found FRL (Remargin) to be more effective than FRL (Reweight.) This may be due to differences in our datasets, or artifacts of my implementation. It should be noted that because of the greater expense of this procedure, it was harder to thoroughly explore its hyperpaparameters, and this is still an interesting area of exploration for me.

## 5 Discussion

I believe that overall my results are quite in line with the original paper's. I found that adversarial training does produce unfair results, both in the improvements to robustness the model receives as well as the degradation of standard error it experiences. I also found that these unequal costs penalize classes that are harder for the model to learn, making it worse at what classifying what it already had trouble with. Finally, I found that the FRL (reweight) approach was

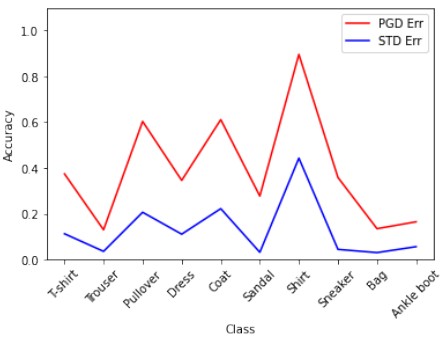 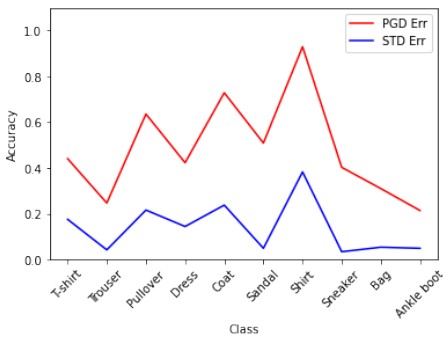

(a) "Vanilla" adversarially-trained model          (b) After the FRL (Remargin) procedure

Figure 3: FRL Remargin was able to slightly improve the standard performance of the hardest class, but decreased the overall robustness of the model

able to mitigate most of the degradation in standard performance for the hardest to learn classes, and to a lesser degree improve the robustness for that class as well as well as the overall robustness.

One weak point of my implementation was in the FRL (Remargin) procedure. I was unable to successfully improve the model's robustness via remargining, though I am not confident that I thoroughly explored the space. It was the most costly procedure I ran, and it ran into its fair share of Colab timeouts, making hyperparameter tuning tricky.

One last experiment I did not have time for was a combination of reweighting and remargining, which Xu et al. suggest is the most effective means increasing adversarial fairness. This is because I wanted positive results in remargining before attempting to combine the two approaches, which I was unfortunately unable to achieve. This is still an open question to pursue.

## 5.1 What was easy

One of the paper's easiest claims to verify was that adversarial training creates the unfair outcomes described above. Even with little experience in adversarial training, we found that with only a bit of effort I could observe this phenomenon myself.

It was also fairly easy to implement Xu et al's FRL algorithms; the remargining and reweighting procedures are very clearly explained in the paper and were straightforward to put into code. One aspect of the paper not discussed in this report is their theoretical analysis, which was also very clear and helped motivate and explain the FRL problem formulation.

## 5.2 What was difficult

As mentioned above, the part I had the most difficulty with was the remargining procedure. It took much longer than anticipated, and its expense made automated hyperparameter searches difficult. Because I was unsuccessful in improving the model's robustness with remargining, I was also hesitant to implement a combined FRL (Reweight) and FRL (Remargin) approach, which the authors suggest might be the most effective result. As mentioned, this is an area in which I am still actively exploring. Hopefully in the future I can replicate their success there too.

## 5.3 Communication with original authors

As mentioned in my summary, I did not have contact with the authors throughout this process. It was only upon drafting my report that I learned it was encouraged to contact the original authors; in the future, I think it would be a great idea to communicate with them sooner. I reached out with a preprint of the report for any feedback or suggestions, but did not hear back.

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
