# OpenReview forum: "[RE] An Implementation of Fair Robust Learning"
_ML_Reproducibility_Challenge/2021/Fall — RC2021 OutstandingPaper_

### Official Review · Reviewer_Xv53 · 2022-02-25
**Accept**

**Rating:** 7
**Confidence:** 4

**Review:**

The authors clearly state the claims and design the corresponding experiments. In order to perform the experiments in low-computation-resources situations, they re-implement the code to use a smaller model and dataset. The reproduction shows the efficacy in a different setting from the original paper.

The authors reproduce the discrepancy given the smaller dataset in traditional adversarial training, and show the efficacy after implementing FRL. The exploration process is complete.

The authors find deviations from the original paper in certain cases. The results provide additional insights.

---

### Meta-Review · Area_Chair_rc9B · 2022-04-07

**Recommendation:** Accept (Outstanding Paper)
**Confidence:** 5

**Metareview:**

The paper presents a strong reproducibility effort - not only the author re-implented the code in publicly available Google Colab, but the author opted for a different model family and a different dataset which is more amenable to training with free Google Colab resources. The fact the paper found most of the results in line with the original paper highlights the strong robust contribution of the original paper involved. The paper perhaps can be further improved with better presentation of the charts in the camera ready version.

This is a very commendable effort, and thus I vote for its acceptance.

---

### Decision · Program_Chairs · 2022-04-09

**Decision:**

Accept (Outstanding Paper)

**Comment:**

Following the recommendation of reviewers and meta-reviewer, the paper is accepted for ML Reproducibility Challenge 2021, and will be published in the upcoming special edition of ReScience Journal.

Additionally, after several rounds of discussion and incorporating recommendations from the Area Chairs and Program Chairs, the report has been granted an **Outstanding Paper Award** due to its exceptional quality of all-round reproducibility effort. Congratulations!